# An integrated method for the leakage fault mode diagnosis and life prediction of the reactor coolant pump

Yinghua Shao[◉]°, Rui Kang[◉]‡, Jingwen Fu‡, Jie Liu[◉]*°

School of Reliability and Systems Engineering, Beihang University, Beijing, China

° These authors contributed equally to this work.
‡ RK and JF also contributed equally to this work.
* liujie805@buaa.edu.cn

**Data Availability Statement:** All relevant data are within the manuscript and its Supporting Information files.

## Abstract

The reactor coolant pump is a key equipment in a nuclear power plant. If the leakage exceeds a certain threshold, it may cause reactor overheating and shutdown. The reactor coolant pump leakage fault usually has two problems: corrosion and scaling. Accurately and efficiently diagnosing the leakage fault mode as early as possible and predicting its remaining useful life (RUL) are important for taking timely maintenance measures. In this paper, an integrated method is proposed. First, the cross-sectional area of the first seal is extracted as a fault indicator. The motivation is that corrosion may enlarge the cross-sectional area, and scaling may reduce the cross-sectional area. Based on the fluid mechanics theory, an integrated model with several uncertain parameters is established among the cross-sectional area, temperature, and leakage at the inlet and outlet of the first seal. In the diagnosing process, a modified change-detection method is proposed to detect the starting point of degradation. Then, the unknown parameters in the previous relation are estimated, and the degrading data before the starting point of degradation are used to diagnose the leakage fault mode. Second, a time-series model of the autoregressive integrated moving average (ARIMA) is established to predict the remaining useful life based on the degrading data after the starting point of degradation. Finally, the leakage degrading data from six reactor coolant pumps of a nuclear power plant is used to perform the leakage fault mode diagnosis and life prediction with degradation point detection error rates not exceeding 4%, fault mode diagnosis correction rates 100% and practical RUL predicting results, which proves that the proposed integrated method is accurate and efficient. The proposed integrated method combines the advantages of both the physical model diagnosis and the data-driven model diagnosis and innovatively make use of the quantity of flow from the output side of the primary pump as the monitoring indicator and the cross-sectional area as the characteristic index together to diagnose the leakage fault mode happened to the seal and predict its RUL, which can meet the needs of actual operation and maintenance to ensure a healthy and stable operation of the pump and prevent unexpected shutdowns of nuclear power plants and serious accidents.

**Funding:** This work was supported by National Natural Science Foundation of China (No. 52005027) (Corresponding author: Jie Liu) and National key Laboratory of Science and Technology on Reliability and Environmental Engineering. The funders have played important role in decision to publish.

## Introduction

Nuclear power plants have been operating in nearly 30 countries to generate electricity, and they account for approximately 17% of the world's total electricity generation [1]. Nuclear energy has become one of the most important sources of energy. Although nuclear power plants in the world have rich operating experience and good safety records, due to the impact of nuclear pollution, the safety of nuclear reactors remains the most concerning issue in the development of nuclear energy [2].

The reactor coolant pump (RCP) is one of the most important equipment in a nuclear power plant (NPP) and is known as the heart of the reactor cooling system, so its safety is crucial [3]. Effective fault diagnosis and health management of the RCP in NPPs during operation are currently a research hotspot. The mechanical seal assembly is the most vulnerable part of an RCP. According to the investigation of the working conditions of the main pump, more than 70% of the failures are caused by the mechanical seal, and the continuous increase in leakage of the primary mechanical seal is one of the main failure phenomena [4]. Therefore, when the leakage of the primary mechanical seal of the nuclear main pump continues to abnormally increase or decrease, an effective method to conduct fault diagnosis and predict its remaining useful life must be found. This method will adjust and control the leakage of the primary mechanical seal in the normal range without affecting the normal operation of the unit. This problem must be fundamentally solved in the field of nuclear power worldwide.

Due to the inherent uncertainty of the faults of the RCP in NPPs, relying solely on the experience and knowledge of operators for fault diagnosis and prediction is difficult to complete all diagnostic tasks. Therefore, in recent years, methods based on data-driven models and fault physical models have become the mainstream for the fault diagnosis and prediction of the RCP in NPPs.

The method based on fault physical models analyzes and predicts the product physical failure mechanisms, which mainly include methods based on cumulative damage [5], those based on warning devices [6], and those based on specific models [7]. The commonly used steps are data preprocessing, fault mode determination, fault mechanism model selection, and model evaluation. The main achievement of methods based on fault physical models is the PRODIAG diagnostic system developed by the Argonne National Laboratory in the United States. PRODIAG combines the basic physical principles of thermal hydraulic processes in its diagnosis. Testing shows that the system can provide a fault set in a relatively short time [8, 9]. Thermal hydraulic analysis is widely used as a kind of fault physical model combined with simulation to carry out the diagnosis and prediction of the nuclear plant: Mohamed Ali has carried out thermal analysis of baffle jetting in fuel rod assembly to compute various flow parameters such as pressure coefficients along different rods, mean and fluctuating forces, Strouhal number, local and averaged Nusselt numbers through large eddy simulations (LES) of flow [10]; Ahmed K. Alkaabi has benchmarked the prediction accuracy of COMSOL Multiphysics in performing thermal-hydraulic analysis of TRIGA (Training, Research, Isotopes, General Atomics) reactors such as the Geological Survey TRIGA Reactor (GSTR) by comparing its predictions with RELAP5 (a widely used code in nuclear thermal-hydraulic analysis) results and experimental data [11]. France developed a model-based online monitoring and diagnosis system for pressurized water reactor NPPs [12], which comprehensively utilizes the mass and energy conservation equations and can diagnose faults related to heat conduction, heat transfer, and power systems [13]. The method based on fault physics models can delve into the essence of the object, and the fault features are closely related to the model parameters. The model has strong interpretability and real-time prediction ability. However, its establishment requires a deep

understanding of the fault evolution mechanism of the object and is not suitable for the prediction modeling of complex, non-linear, and uncertain objects.

Data-driven model-based methods utilize the historical or current data of an object under certain functional constraints to establish a type that can approximate the implicit mapping mechanism between object data and lifespan for prediction [14]. The main steps of data-driven methods are data preprocessing, feature extraction, feature selection, model selection, and model evaluation [15–17]. Data-driven model-based methods mainly includes methods based on statistical regression [18], similarity [19], and the stochastic process [5, 20]. Zhang first used the bootstrap method to resample the raw data, subsequently trained neural networks with each subset of the obtained data, and finally comprehensively processed the diagnostic results of these neural networks [21]. Zhao et al. used the Back-Propagation artificial neural network to diagnose accidents in the AP1000 NPP [22]. Mao Wei et al. combined support vector machines with wavelet decomposition for the fault diagnosis of main pumps in NPPs [23]. The University of Tennessee in the United States designed a fault diagnosis method based on a combination of principal component analysis (PCA) and rule-based reasoning [24]. Professor Gofuku from Kyoto University in Japan used a linear preserving projection algorithm for fault mode classification [25]. Park et al. from the Korean Academy of Atomic Energy Sciences used PCA with distance functions for a typical fault diagnosis in secondary circuit systems [26]. Xie C. et al. conducted many studies on the application of data fusion technology to the fault diagnosis of NPPs [27]. Causality is also used to carry out diagnosis of complex mechatronic systems. Zheng Shuwen et al. carry out fault detection in complex mechatronic systems by a hierarchical graph convolution attention network based on causal paths [28]. Liu Jie et al. proposed a novel method for online fatigue crack size evaluations using acoustic emission signals by leveraging the advanced causal network and graph neural networks techniques [29]. Xu Yubo et al. proposed causality-based principal component analysis methods for condition modeling of mechatronic systems [30]. Data-driven tools such as time-series vision transformer [31] are widely used in fault diagnosis. This type of method does not require an understanding of the fault mechanism or empirical knowledge of the object. It only uses various data processing and analysis methods to mine hidden information, which can compensate for the shortcomings of data analysis methods based on fault physical models. The fault diagnosis accuracy and prediction accuracy are high, but because the fault mechanism was not explained, the model interpretability is not strong, and long-term analysis and prediction cannot be performed.

In general, both methods show limitation when applied to real-time fault diagnosis in the power plant industry. To solve the limitations of both methods, an integrated method is proposed in this paper to perform the fault diagnosis and life prediction for the RCP of NPPs, which can monitor the leakage during the operation of the nuclear main pump. To satisfy the requirements of the actual operation and maintenance, this method attempts to combine the advantages of the physical model diagnosis and data-driven model diagnosis. The quantity of flow from the output side of the primary pump is the monitoring indicator, the cross-sectional area is the characteristic index, and they are used together to diagnose the pattern of failure of the seal and predict its remaining useful life. Thus, proper measures can be taken when the leakage quantity is abnormal to ensure a healthy and stable operation of the pump and prevent unexpected shutdowns of NPPs and serious accidents.

The remainder of this manuscript is structured as follows. Section 2 ('Structure of the Integrated Method') introduces the structure of this integrated method and details all steps. Section 3 ('Case Study') presents a case study to verify the effectiveness of the method. Finally, a conclusion is made to evaluate the applicability and improvement direction of the method.

## Structure of the integrated method

As shown in Fig 1, the integrated method can be divided into three major steps. The first step is called Health Monitoring. In this step, the degradation point [32, 33] of the leakage data is found as a dividing point, for which all data before the degradation point can be considered normal data, and all data after the degradation point can be considered abnormal data. In this case, we assume that before the degradation point, the primary seal maintains normal operation, and no fault mode occurs, so the data can be used to build the model and optimize the parameters. On the contrary, we assume that after the degradation point, failure occurs to the primary seal, so that all data after the degradation point are used to diagnose the fault mode of the primary seal.

The second step is called fault mode diagnosis. The process of fault diagnosis shown in Fig 2 is divided into four main steps:

1.  Physical model construction. We must construct the physical relationship between detection value and characteristic value to analyze the fault model of the primary sealing device through the existing detection value data.

2.  Parameter magnitude analysis and optimization. In the obtained model, due to unknown parameters, we must optimize the parameters in a data-driven manner. Although the model is based on physics instead of being purely data-driven, each parameter has its actual physical meaning, so the magnitude of the parameter can be roughly deduced before the optimization process to make the optimization parameter more specific and reduce the amount of calculation. Then, an optimization program to minimize error compared to actual leakage data based on Python [34] is used to optimize the unknown parameters in the equation obtained in step (1).

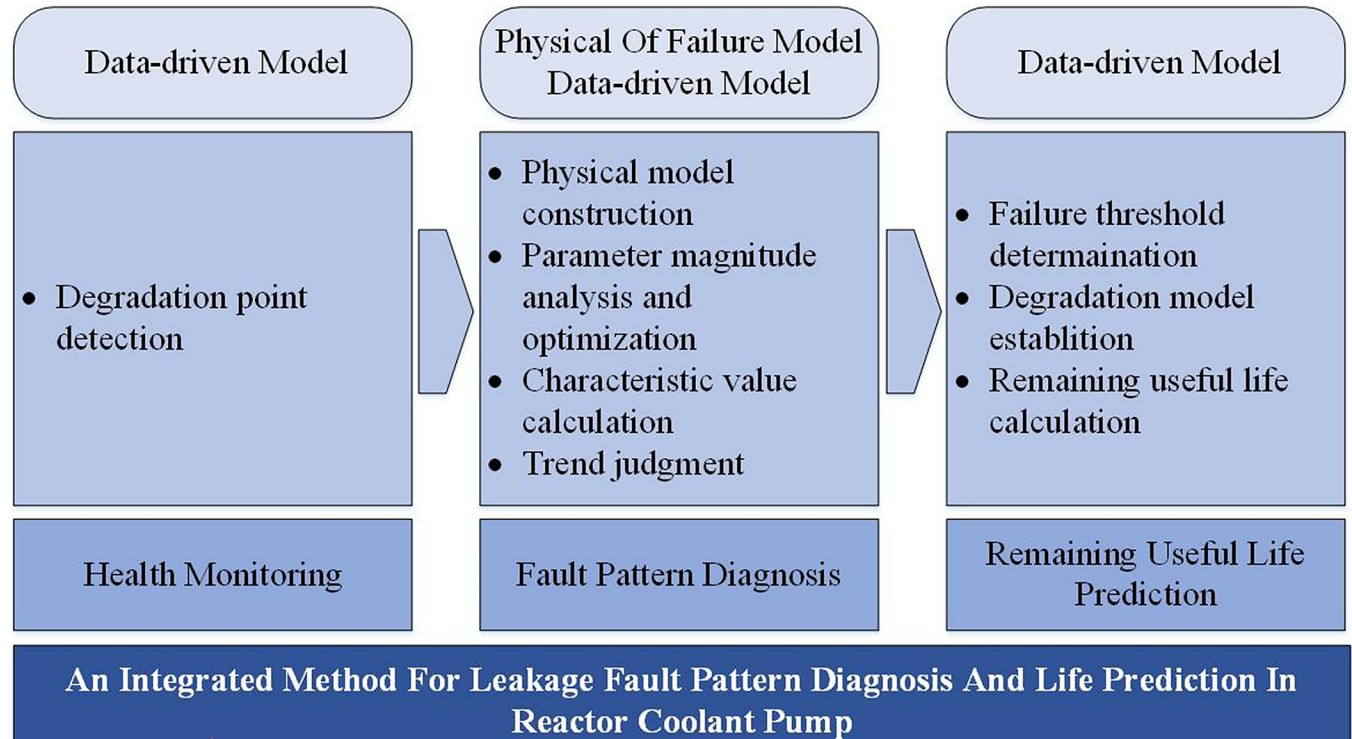

**Fig 1. Structure of the integrated method.**

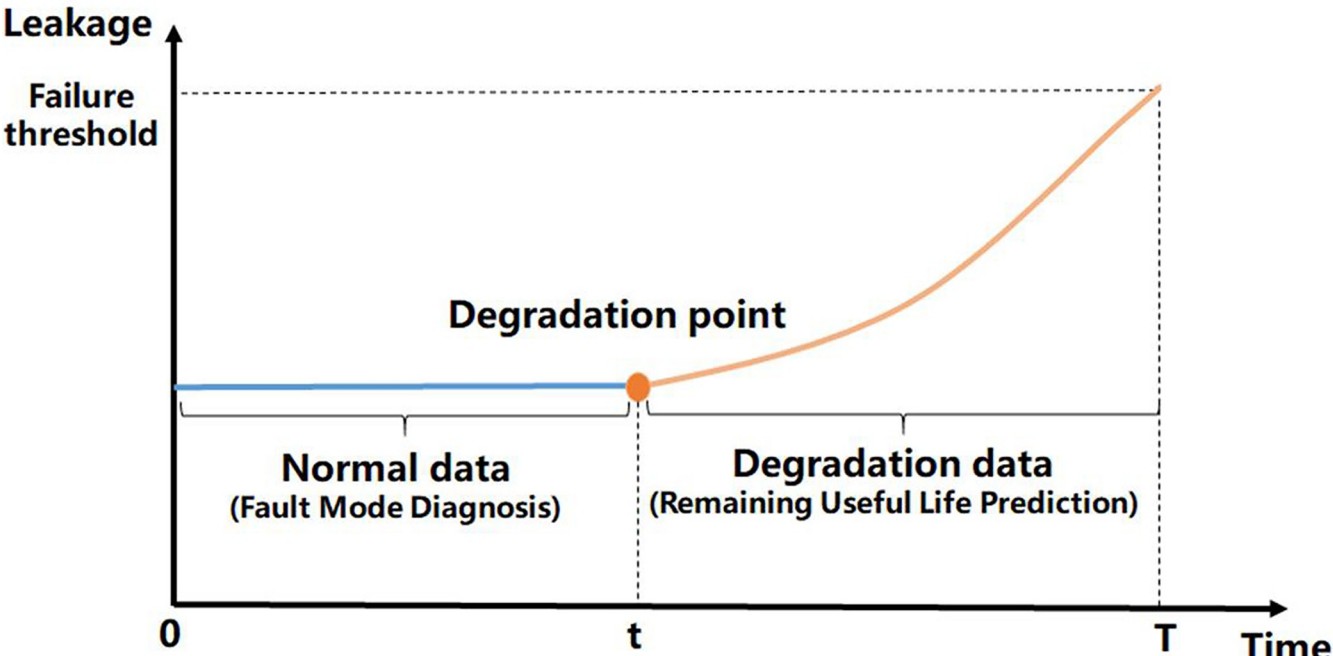

**Fig 2. Process of the integrated method.**

3. Characteristic value calculation. Introducing the parameters in the previous step and the known detection values into the equation obtained in step (1) to find the trend of the characteristic value.

4. Trend judgment. Some methods are used to determine the trend of the obtained characteristic value curve to diagnose the fault type of the primary sealing device.

The third step is to predict the remaining useful life. First, the failure threshold of the monitored data is determined. Second, the degradation process is modeled with the monitoring data after the degradation point. Third, the remaining useful life can be obtained by the failure threshold and degradation model.

## Health monitoring

Degradation point detection is one of the most important works in health monitoring. The degradation point, which is also called the change point, is the beginning of the change in monitoring status and the beginning of system failure. Detecting the degradation point is the challenge in fault diagnosis and prediction. The N-point average value detection method is proposed in this paper to detect the degradation point.

Record all data points as $Q_1, Q_2, \ldots, Q_i, \ldots, Q_N$ for the N-point average method; for any integer ($i \geq n+1$), let

$$\overline{Q}_{i, n} = \frac{x_i + x_{i+1} + \ldots + x_{i+n-1}}{n},$$

$$\overline{Q}_{i-1, n} = \frac{x_{i-n} + x_{i-n+1} + \ldots + x_{i-1}}{n}.$$

The requirement to be met is as follows:

$$\bar{Q}_{i,n} > a\bar{Q}_{i-1,n}.$$ 

(1)

In addition, the following requirement should be met:

$$Q_i > b\bar{Q}_i,$$ 

(2)

where *a* and *b* are two coefficients that can be determined or fitted based on the actual needs of health monitoring. Then, this point is the point of leakage change, which is also known as the starting point of shaft seal degradation.

## Fault mode diagnosis

After the degradation point has been detected, fault mode diagnosis can be performed with normal data before the degradation point.

**Physical model construction.** The two fault modes for the mechanical seal are corrosion [35] and abrasion [36]. The cross-sectional area of the seal will increase when corrosion occurs and decrease when abrasion occurs [37]. Thus, we can use the cross-sectional area of the low channel as the fault characteristic quantity to indicate the fault mode of the mechanical seal to help the fault diagnosis when the seal leakage is abnormal.

In this article, we simplify the research on the primary mechanical seal into the model in Fig 3. We simplified the coolant flowing through the sealing device into an ideal fluid model [38]. The ideal fluid model refers to a fluid model that ignores the effects of diffusion, viscosity,

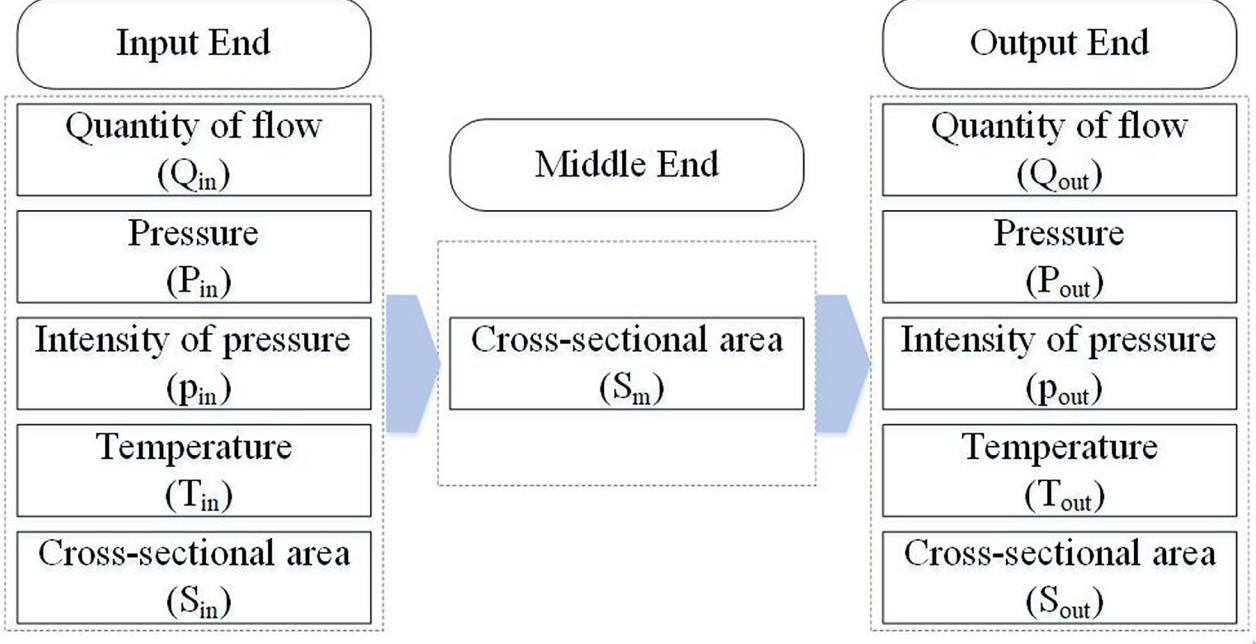

**Fig 3. Simplified model of the primary seal of a nuclear pump.**

and heat conduction and is called an inviscid fluid [39]. The ideal fluid model can be divided into a compressible ideal fluid model [40] and an incompressible ideal fluid model [41] according to the compressibility. Here, we choose the incompressible fluid model.

According to Bernoulli's equation [42], for an ideal fluid to stably flow, at any point in the same flow tube, the sum of the kinetic energy and potential energy per unit volume of the fluid and the pressure at that point is constant [43], as shown in Fig 4.

$$p_1 + \frac{1}{2}\rho v_1^2 + \rho g h_1 = p_2 + \frac{1}{2}\rho v_2^2 + \rho g h_2, \tag{3}$$

$$p + \frac{1}{2}\rho v^2 + \rho g h = C, \tag{4}$$

where $p$ is the pressure at a certain point in the fluid, $v$ is the flow velocity of the fluid at that point, $\rho$ is the fluid density, $g$ is the acceleration of gravity, $h$ is the height of the point, and $C$ is a constant number. In this study, the input end and output end of the shaft seal are on the same horizontal plane, so the potential energy has negligible influence on the fluid, and the following is obtained:

$$p + \frac{1}{2}\rho v^2 = C. \tag{5}$$

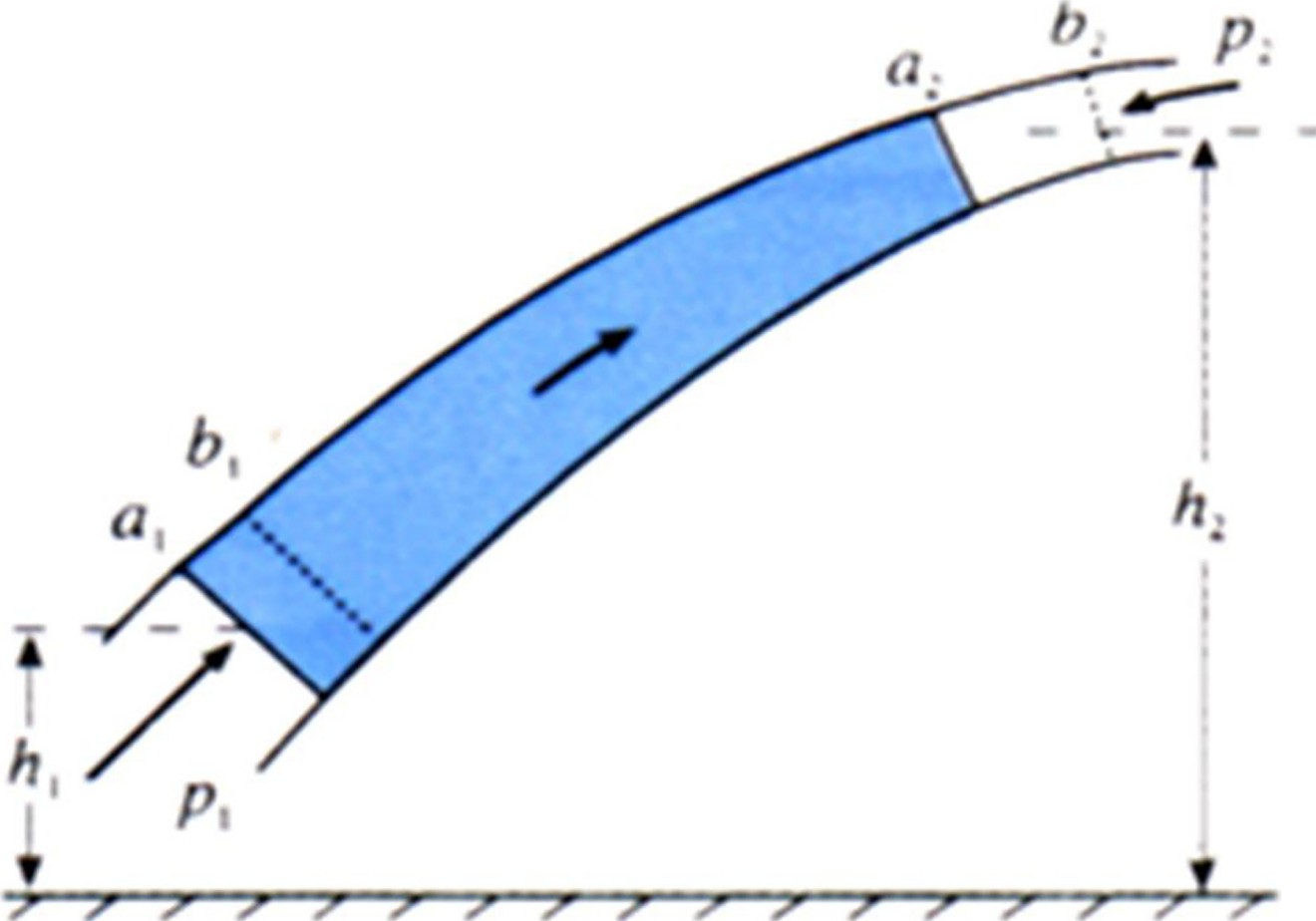

**Fig 4. Bernoulli's equation of hydrodynamics.**

Since the flow rate is equal to the fluid velocity multiplied by the cross-sectional area, the following can be obtained:

$$Q = v \cdot S, \tag{6}$$

where $Q$ is the flow rate of the fluid at this point, $v$ is the velocity of the fluid at this point, and $S$ is the cross-sectional area of the point.

According to the stable flow formula, the following can be obtained:

$$Q = \mu S \cdot \sqrt{\frac{p_2 - p_1}{\rho}}, \tag{7}$$

where $Q$ is the quantity of the fluid, $\mu$ is a constant coefficient, $S$ is the flow area, $p_1$ is the pressure value at the front end of the fluid, $p_2$ is the pressure value at the back end of the fluid, and $\rho$ is the density of the fluid.

At the input end of the nuclear main pump, we have

$$p_{in} + \rho \cdot \frac{v_{in}^2}{2} = C_{in}, \tag{8}$$

where $p_{in}$ is the pressure at the pipeline input end of the nuclear main pump primary seal device, $\rho$ is the fluid density in the pipeline of the nuclear main pump primary seal device, $v_{in}$ is the fluid velocity at the pipeline input end of the nuclear main pump primary seal device, and $C_{in}$ is a constant.

At the output end of the nuclear main pump, we have

$$p_{out} + \rho \cdot \frac{v_{out}^2}{2} = C_{out}, \tag{9}$$

where $p_{out}$ is the pressure at the pipeline output end of the nuclear main pump primary seal device, $\rho$ is the fluid density in the pipeline of the nuclear main pump primary seal device, $v_{out}$ is the fluid velocity at the pipeline output end of the nuclear main pump primary seal device, and $C_{out}$ is a constant.

Introducing Eqs (6), (8), and (9) into Eq (7) yields

$$S_m = \frac{C_1 Q_m}{\sqrt{C_2 + C_3 Q_{in}^2 - C_4 Q_{out}^2}}. \tag{10}$$

Parameters $C_1$, $C_2$, $C_3$, and $C_4$ are introduced, where

$$C_1 = \frac{1}{\mu}, \ C_2 = \frac{C_{out} - C_{in}}{\rho}, \ C_3 = \frac{1}{2S_{in}^2}, \text{ and } C_4 = \frac{1}{2S_{out}^2}.$$

In Eq (10), $S_m$ is the cross-sectional area of the pipeline of the primary seal device of the nuclear main pump, $Q_m$ is the total quantity of the coolant that flows through the pipeline, $\mu$ is a constant coefficient, $C_{out}$ is a constant, and $C_{in}$ is a constant. $Q_{in}$ is the quantity of coolant flowing through the input end of the nuclear main pump primary seal device, and $Q_{out}$ is the quantity of coolant flowing through the output end of the nuclear main pump primary seal device.

In the actual working conditions of an RCP, the temperature changes from the input end to the output end. Thus, the temperature changes must be considered in the model establishment process. From Peng-Robinson equation [44], the pressure is approximately proportional to the

temperature:

$$p = kT, \tag{11}$$

where $p$ is the pressure of the fluid, $k$ is a constant coefficient, and T is the temperature of the fluid. Using Eqs (10) and (11), the model that considers the temperature change conditions can be derived as follows:

$$S_m = \frac{Q_{out}}{\sqrt{C_1 \cdot T_{in} - C_2 \cdot T_{out} + C_3 \cdot T_{out} \cdot Q_{out}^2 - C_4 \cdot T_{in} \cdot Q_{in}^2}} \tag{12}$$

where $T_{in}$ is the temperature of the input end, and $T_{out}$ is the temperature of the output end. Fig 5 shows the primary seal fault diagnosis model of the nuclear main pump.

According to the existing leakage data of six sets of pumps, we can use Python to optimize the parameters to obtain specific values. $S_m$ can further be calculated by introducing $T_{in}$, $T_{out}$, $Q_{in}$, and $Q_{out}$ to Eq (12). When $S_m$ shows a growing trend and exceeds the set threshold, the fault mode of the nuclear main pump is corrosion. When $S_m$ shows a decreasing trend and exceeds the set threshold, the fault mode of the nuclear main pump is scaling.

**Parameter magnitude analysis and optimization.** According to this derivation process, after the temperature standardization constants $T_{ins}$ and $T_{outs}$ have been introduced, the four

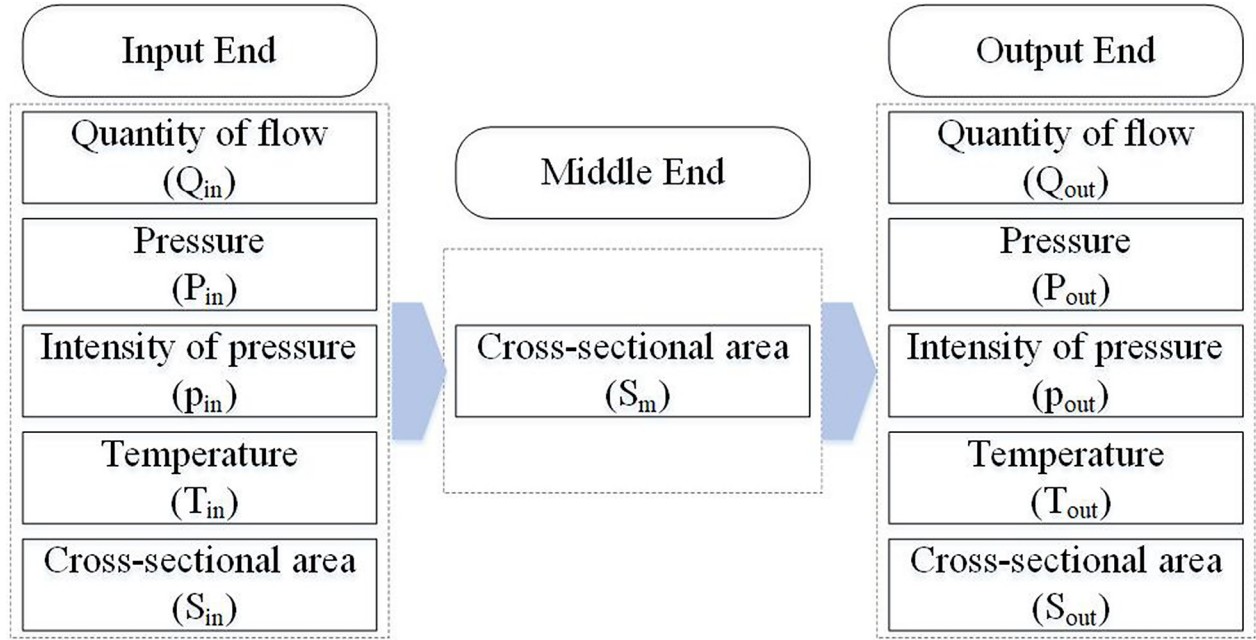

**Fig 5. Simplified model of the primary seal fault diagnosis.**

parameters in Eq (12) can be calculated as follows:

$$C_1 = \mu \cdot \sqrt{\frac{k}{\rho}} \cdot \frac{C_{in}}{T_{ins}}, \tag{13}$$

$$C_2 = \mu \cdot \sqrt{\frac{k}{\rho}} \cdot \frac{C_{out}}{T_{outs}}, \tag{14}$$

$$C_3 = \mu \cdot \sqrt{\frac{k}{\rho}} \cdot \frac{\rho}{2T_{outs} \cdot S_{out}{}^2}, \tag{15}$$

$$C_4 = \mu \cdot \sqrt{\frac{k}{\rho}} \cdot \frac{\rho}{2T_{ins} \cdot S_{in}{}^2}. \tag{16}$$

According to Bernoulli's theorem, the constant coefficients at the output and input are roughly identical, i.e.,

$$p_{in} + \frac{1}{2}\rho v_{in}{}^2 \approx p_{out} + \frac{1}{2}\rho v_{out}{}^2. \tag{17}$$

Thus, we obtain $C_{in} \approx C_{out}$.

Since $T_{ins} \approx T_{outs}$, we obtain $C_1 \approx C_2$.

Because the cross-sectional area is larger at the input end than at the output end. $S_{in}$ is far larger than $S_{out}$.

Taking the experience value $S_{in} = 10S_{out}$, we obtain that $C_3$ is far larger than $C_4$.

Let $C_4 = 1$; then, we can obtain $C_3 = 100$, and the relationship between $C_2$ and $C_3$ can be expressed as follows:

$$\frac{C_2}{C_3} = \frac{2C_{out}S_{out}{}^2}{\rho} = \frac{2C_{out}}{\rho}\left(\frac{Q_{out}}{v_{out}}\right)^2. \tag{18}$$

In the actual fault diagnosis process, $C_1$、 $C_2$、 $C_3$、 $C_4$ are input to Python for parameter optimization to obtain specific values using the on-site monitoring data of mechanical seal leakage of RCP.

**Characteristic value calculation.** In this step, $C_1$、 $C_2$、 $C_3$、 $C_4$ calculated from the last step and $T_{in}, T_{out}, Q_{in}, Q_{out}$ from on-site detection are applies to Eq (12) to obtain a specific value of $S_m$. Then a curve of $S_m$ trend is drawn.

**Trend judgment.** The data after the degradation point are used for trend judgment to analyze the degradation mode. In this article, the Cox-Stuart detection method is used to monitor the trend of the curve [45]. The principle of the Cox-Stuart monitoring method is to directly consider the changing trend of the data. If the data have an upward trend, the value of the next data point is significantly larger than the value of the previous data point. On the contrary, if the data have a downward trend, the value of the previous data point is significantly smaller than the value of the next data point. This method uses the positive or negative difference of different data points in the two periods before and after to determine the overall trend of the data. The algorithm steps are as follows:

For $n$ data points, take the $i$-th data point $x_i$ and the $(i+c)$-th data point $x_{i+c}$ from a pair of arrays $(x_i, x_{i+c})$. For integer $n$, if $n$ is even, $c = n/2$; if $n$ is odd, $c = (n+1)/2$. When $n$ is an even number, there are $n'=c$ pairs of data. When $n$ is odd, there are $n'=c-1$ pairs of data. The sign of

$D_i = x_i - x_{i+c}$ is used to measure the increase or decrease of the data sequence. $S_+$ is calculated as the number of positive $D_i$, and $S_-$ is the number of negative $D_i$. When $S_+ \geq S_-$, the data sequence shows a downward trend. When $S_+ < S_-$, the data sequence shows an upward trend. Under the zero assumption without a trend, $D_i$ should follow a binomial distribution $b(n',0.5)$.

Based on the trend of change, if $S_m$ shows an upward trend, the fault mode is corrosion. If $S_m$ shows a downward trend, the fault mode is scaling.

## Remaining useful life prediction

After the fault mode is diagnosed, remaining useful life prediction can be performed with the abnormal data after the degradation point.

**Failure threshold determination.** According to the primary sealing principle of the main pump in the NPP, the primary seal is the hydrostatic type, which controls leakage suspended by the liquid film stiffness. The primary seal has a dynamic ring and a static ring. During the working process, the sealing surfaces of the dynamic and static rings are always lubricated by the full fluid mode, which can effectively avoid end face wear. The shape of the dynamic and static ring end faces, the dynamic characteristics of the sealing ring, and the medium control the opening of the end faces and end clearance during stable operation. The leakage amount of the primary seal is the monitoring indicator of the working state of the primary seal, and a stable leakage amount indicates that the sealing working state is normal. In actual work, the leakage amount of the primary seal is easily affected by the pressure, temperature, and other parameters, which cause abnormal situations such as a large or small leakage amount. If the leakage amount exceeds a certain range, it will cause seal failure [46].

**Degradation model establishment.** Since the leakage monitoring of NPPs generally involves data collection at fixed time intervals, the time-series model can be used to model the leakage degradation process of NPPs. In this paper, we introduce the most commonly used autoregressive integrated moving average (ARIMA) model [47, 48].

The ARIMA model is one of the time-series prediction and analysis methods. In ARIMA ($p,d,q$), AR means "autoregressive"; MA refers to the "moving average [49]. Its general mathematical expression is as follows:

$$y_t = \mu + \sum_{i=1}^{p} \phi_i \cdot y_{t-i} + \sum_{j=1}^{q} \theta_j \cdot e_{t-j} \tag{19}$$

where $\phi$ is the coefficient of AR; $\theta$ is the coefficient of MA; $p$ is the lag number of the time-series data in the prediction model, which is also known as the AR term; $q$ is the lag number (lags) of the prediction error in the prediction model, which is also known as the MA term; and d represents the time-series data that must be differentiated by several orders to be stable, which is also called an integrated item.

As shown in Fig 6, there are six steps to establish the ARIMA model: acquisition of time series, pre-processing of time series, model identification, model order determination, parameter estimation, and model validation.

**Remaining useful life analysis.** When the leakage rate of an NPP is abnormal, the leakage rate degradation model can be established based on the leakage rate data after the degradation point has been found at this time to predict the leakage rate at a future time. When the leakage rate predicted by the leakage rate degradation model of the NPP exceeds the upper or lower limit of the leakage rate for the first time, the difference between the time when the leakage rate threshold is exceeded and the current time is the remaining useful life of the main pump of the NPP at the current time.

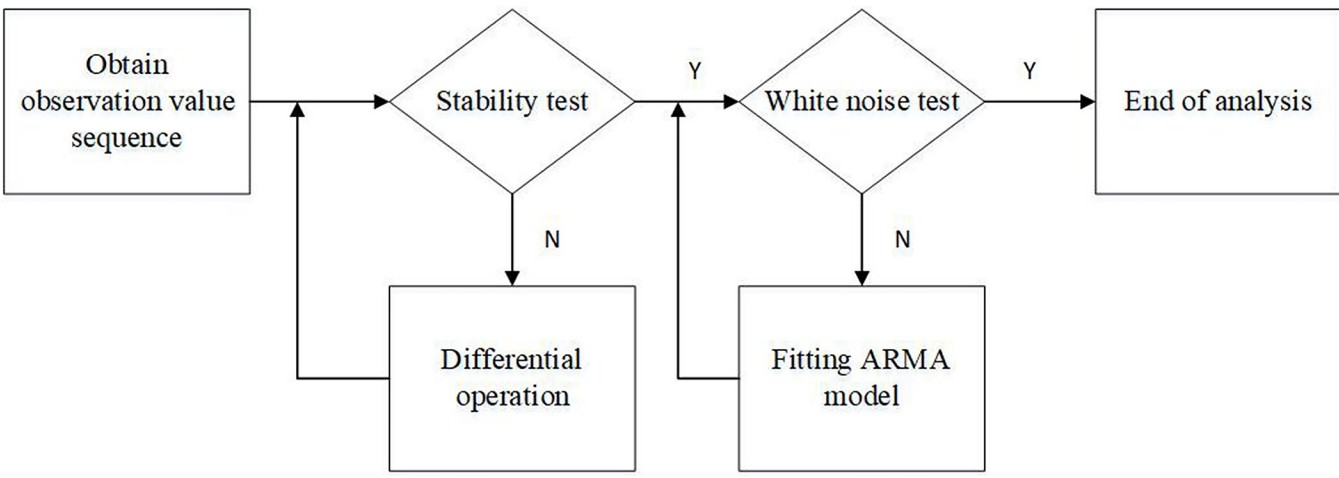

**Fig 6. Procedure to establish the ARIMA model.**

## Case study

In this part, case studies are considered to elaborate on the application of the proposed method for leakage fault mode diagnosis in RCPs.

### Data description

In this case, the monitoring data of the primary seal leakage of six nuclear main pumps in an NPP reactor were selected. For each nuclear main pump, according to a fixed time interval of 4.248 h, four indicators in its working process were monitored and recorded in time sequence. Figs 7 and 8 show the temperature at the input end of the primary mechanical seal device of the nuclear main pump $T_{in}$, temperature at the output end $T_{out}$, leakage at the input end $Q_{in}$, and leakage at the output end $Q_{out}$. In Fig 9, for each nuclear main pump, the corresponding real degradation point and its real fault mode are provided in its technical report. Table 1 shows the references for the degradation point detection in this study and accuracy of the fault diagnosis results.

### Health monitoring

According to the N-point average value detection method, let $a = \frac{\overline{Q_{i,n}}}{Q_{i-1,n}}, b = \frac{Q_i}{Q_i}$.

 Take $n$ = 10,15,20,25,30,35,40,45,50,55,60, traverse $a$ and $b$, and compare the detection results of the detection method with the true values of the six pump degradation points. Table 2 shows the error rate results. In this method, there are three parameters $a, b$ and $n$, which need to be optimized. The constraint objective of optimization is to minimize the error between the obtained degradation point and the true degradation point in Table 1. Subsequently, this article uses Python programs to optimize and finally obtain the optimal value of the three parameters, which is $n$ = 55, $a$ = 1.122, $b$ = 1.126 shown in Fig 9, and at this point, the minimum error rate is 11.7% shown in Table 2. Used the 55-piont average value detection method with parameters $a$ = 1.122, $b$ = 1.126, the degradation points of six pumps can be determined in Table 3 with error rates not exceeding 4% which is proved to be effective.

### Fault mode diagnosis

 **Parameters of the physical model optimization.** Based on the physics of failure model (12), according to the estimated magnitude range of parameter $C$, the parameters are

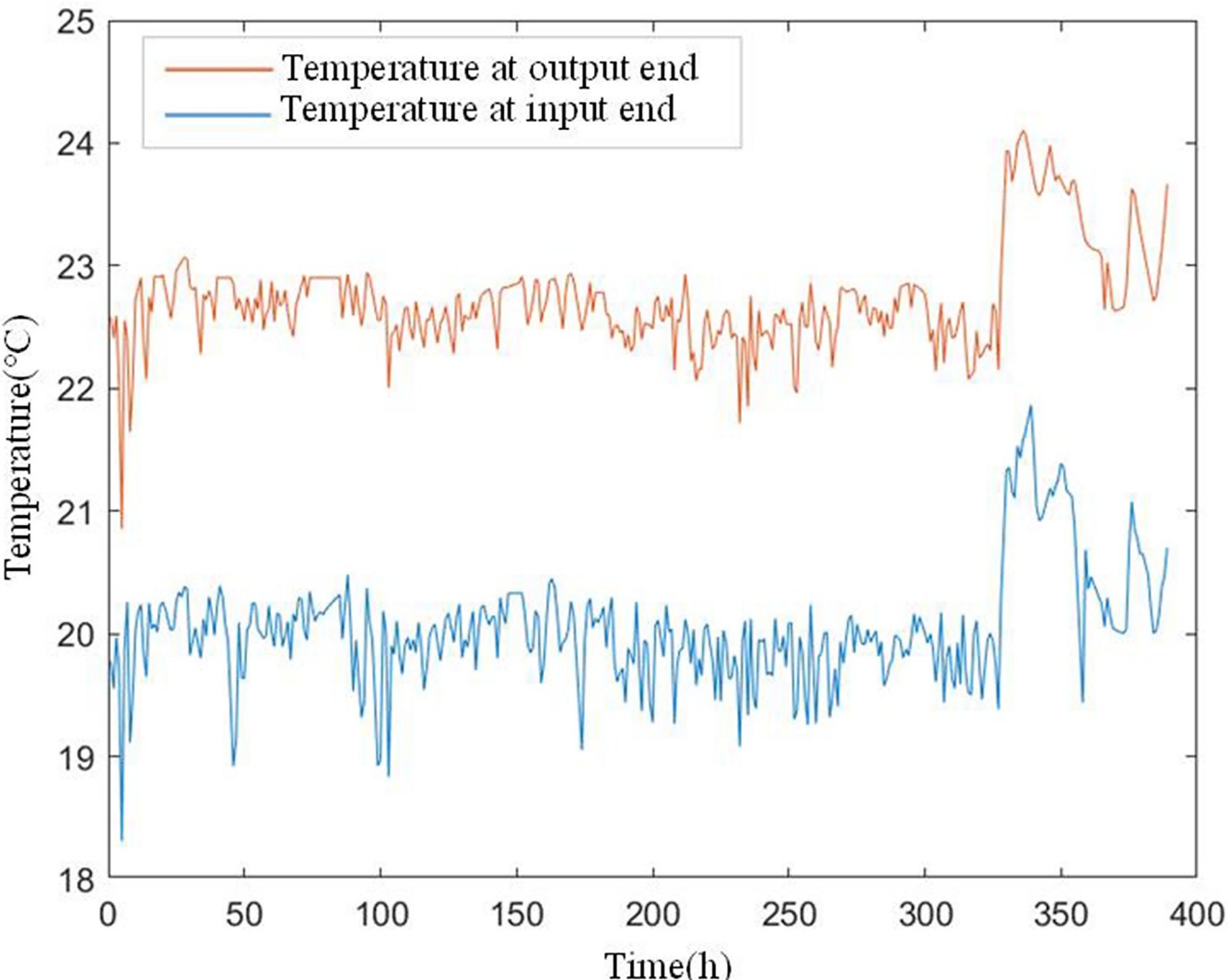

**Fig 7. Temperature at input and output end.**

introduced into Python for optimization. Table 4 shows the actual parameters of six pumps. Fig 10 shows the curve for $S_m$.

**Trend detection.** The trend detection of the curves of $S_m$ changing with time in Fig 10 is carried out using the Cox-Stuart detection method. Under Cox-Stuart detection method, when $S_+ < S_-$, the trend of $S_m$ is rising; while $S_+ > S_-$ the trend of $S_m$ is Descending. Table 5 shows the trend judgement results using the Cox-Stuart detection method.

**Fault diagnosis results.** As shown in Table 5, the cross-sectional areas of No. 1, No. 2, and No. 3 pumps have an upward trend, and the fault mode is determined to be corrosion. The cross-sectional areas of No. 4, No. 5, and No. 6 pumps have a downward trend, and the fault mode is determined to be scaling.

## Remaining useful life prediction

**Determination of failure threshold.** During the operation of the main pump of an NPP, to ensure the operation safety of the NPP, maintenance measures are usually not taken until

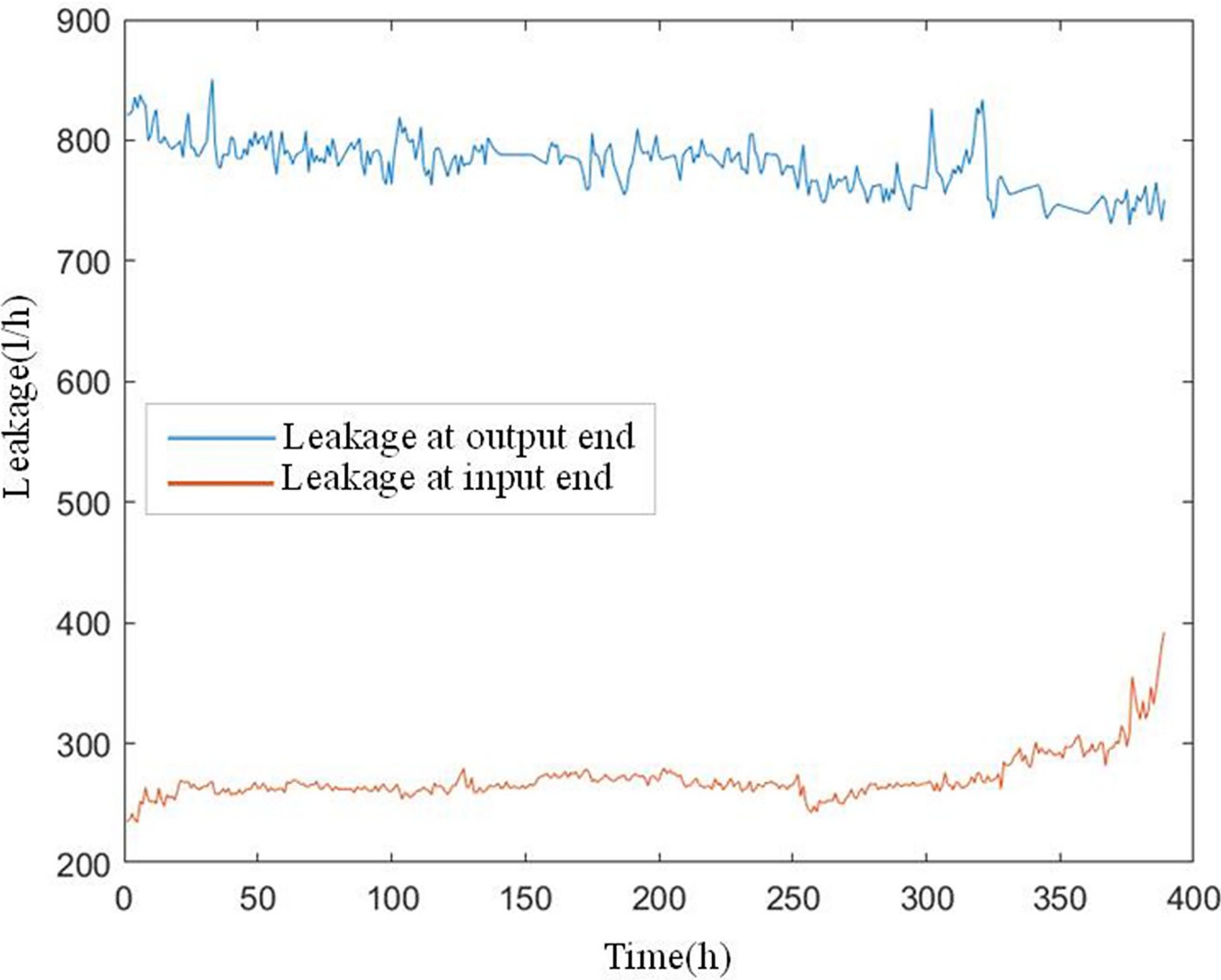

**Fig 8. Leakage at input and output end.**

the leakage of the NPP reaches the failure threshold (upper limit: 1200 l/h, lower limit: 60 l/h). However, when the leakage of the NPP exceeds a certain limit, maintenance measures must be taken. In this case, we take 120% of the normal value as the threshold value to predict the remaining service life of the NPP. Table 6 shows the normal value and service threshold of the leakage volume of the six main nuclear pumps.

**Degradation process modeling based on the ARIMA model.** According to the leakage data after the degradation point of six nuclear main pumps, we established a prediction model based on ARIMA using Python programming, as shown in Figs 11 and 12. Fig 11 shows the ARIMA model for short-term prediction. It forecasts six points with high prediction accuracy. Fig 12 shows the ARIMA model for long-term prediction. It can predict the long-term change trend by predicting 1000 points. According to the "mse" value, the ARIMA model can model the degradation process with better leakage data, whether it is short-term prediction or long-term prediction.

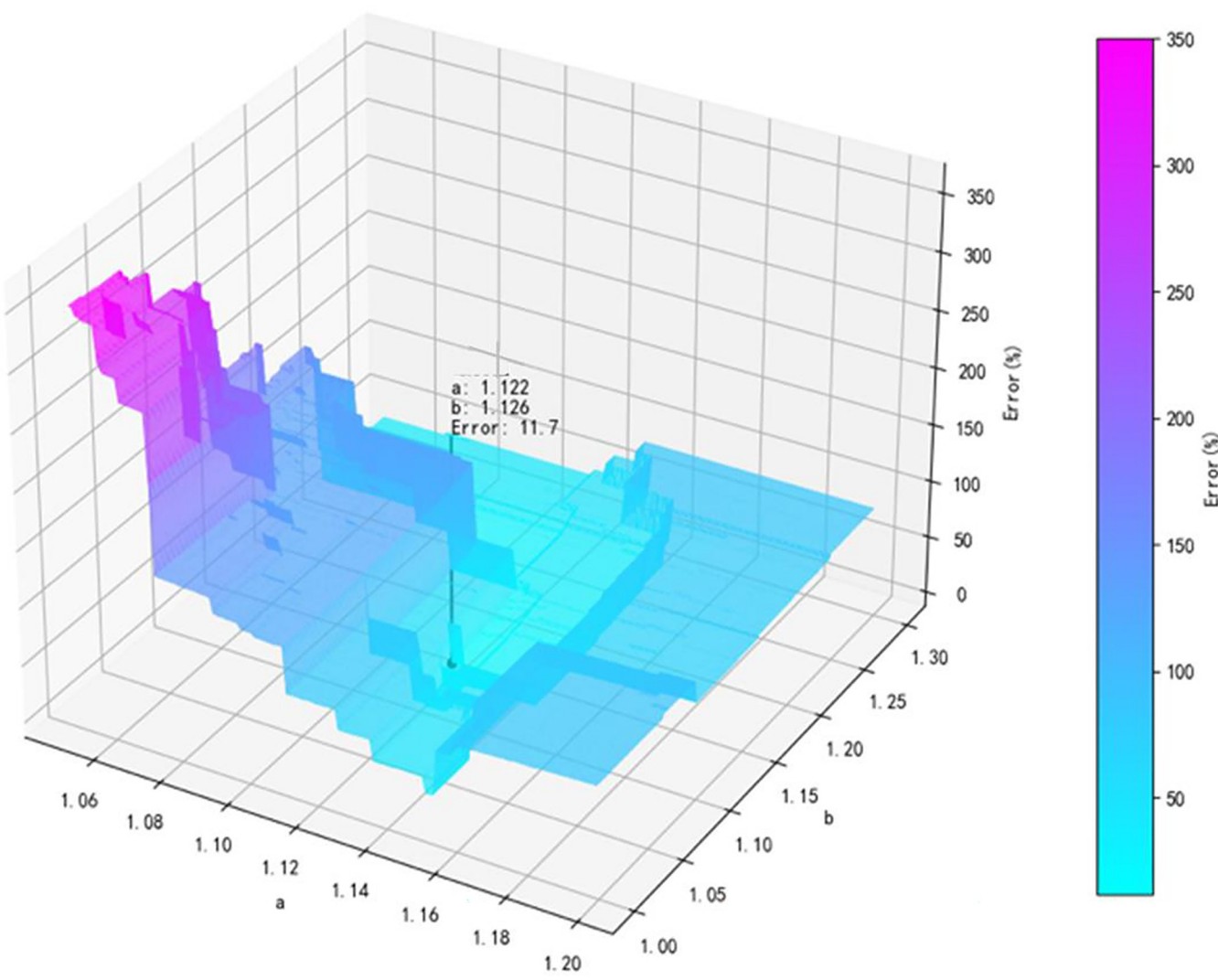

**Fig 9. When n = 55, solve the optimal value image.**

**Remaining useful life prediction.** According to the ARIMA model, the corresponding failure point when the leakage threshold is reached for the first time can be obtained. The time difference between failure point and observation point is the remaining service life. The solution results of the remaining service life of the primary seal of the main pump in six NPPs are shown in Table 7.

**Table 1. Working status of the primary seals of six nuclear main pumps of a nuclear power plant reactor.**

| Pump No. | Total monitoring data | Leakage degradation point | Fault mode |
|---|---|---|---|
| 1 | 2121 | 1900 | Corrosion |
| 2 | 2121 | 2000 | Corrosion |
| 3 | 968 | 650 | Corrosion |
| 4 | 1395 | 950 | Scaling |
| 5 | 566 | 450 | Scaling |
| 6 | 566 | 450 | Scaling |

**Table 2. Optimal solution of a and b for different values of n.**

| n | a | b | Error rate |
|---|---|---|---|
| 10 | 1.143 | 1.096 | 84.0% |
| 15 | 1.161 | 1.096 | 81.8% |
| 20 | 1.107 | 1.257 | 80.3% |
| 25 | 1.074 | 1.284 | 51.5% |
| 30 | 1.079 | 1.252 | 44.7% |
| 35 | 1.115 | 1.176 | 26.5% |
| 40 | 1.101 | 1.194 | 22.1% |
| 45 | 1.129 | 1.153 | 16.4% |
| 50 | 1.122 | 1.103 | 11.8% |
| 55 | 1.122 | 1.126 | 11.7% |
| 60 | 1.128 | 1.103 | 11.9% |

**Table 3. Detection error rate of each pump when n = 55.**

| Pump No. | Actual value | Average detection value of n points | Error rate |
|---|---|---|---|
| 1 | 1900 | 1969 | 3.6% |
| 2 | 2000 | 2037 | 1.8% |
| 3 | 650 | 661 | 1.7% |
| 4 | 950 | 945 | 0.5% |
| 5 | 450 | 458 | 1.8% |
| 6 | 450 | 460 | 2.2% |

## Conclusions

This paper proposes a new method for analyzing the leakage pattern of RCPs, which combines the advantages of the physical model and data-driven method. With the degradation point detection error rates not exceeding 4%, fault mode diagnosis correction rates 100% and practical RUL predicting results, the integrated method proposed is proved to be accurate, efficient and practical. This method is simple to implement and has low calculation complexity. Moreover, it reveals the relationship between detectable value $Q_{out}$ and undetectable characteristic value $S_m$, so that the changing trend of $S_m$ can be inferred and used for further maintenance measures.

However, this method has certain limitations. First, in the physical model derivation, we assume that the model is an ideal fluid with horizontal movement. However, in practical engineering applications, there can be problems such as height difference or heat exchange, which

**Table 4. Parameter optimization results.**

| Pump No. | $C_1$ | $C_2$ | $C_3$ | $C_4$ |
|---|---|---|---|---|
| 1 | $9.3439 \times 10^{-5}$ | $7.4205 \times 10^{-5}$ | 119.9993 | 1 |
| 2 | $4.4452 \times 10^{-5}$ | $7.1026 \times 10^{-5}$ | 119.9998 | 1 |
| 3 | $3.6760 \times 10^{-5}$ | $1.0548 \times 10^{-5}$ | 119.9990 | 1 |
| 4 | $1.0017 \times^{-4}$ | $2.1146 \times 10^{-5}$ | 119.9989 | 1 |
| 5 | $1.2932 \times 10^{-4}$ | $2.5172 \times 10^{-5}$ | 119.9982 | 1 |
| 6 | $5.5493 \times 10^{-5}$ | $9.4066 \times 10^{-6}$ | 119.9994 | 1 |

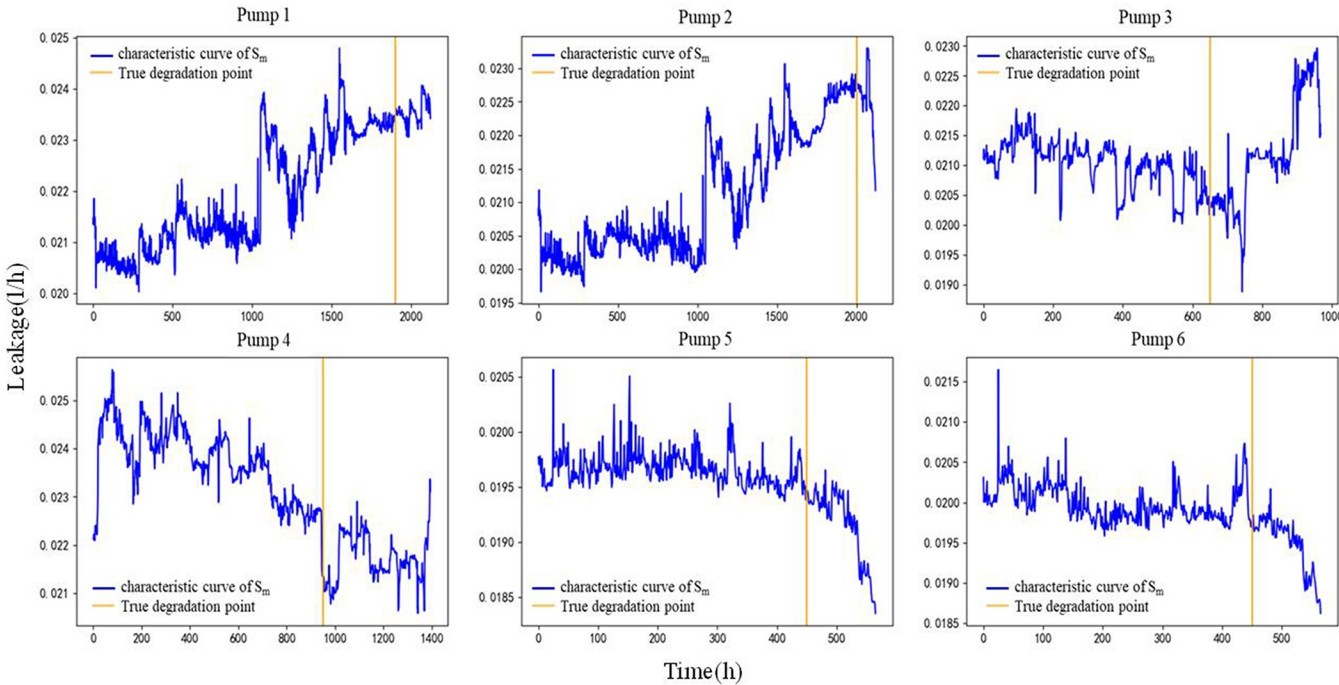

**Fig 10. Curve obtained after parameter optimization.**

**Table 5. Test results of curve change trend judgement.**

| Pump No. | Cox-Stuart detection | Trend judgement results | Fault diagnosis results |
|---|---|---|---|
| 1 | $S_+ < S_-$ | Ascend | Corrosion |
| 2 | $S_+ < S_-$ | Ascend | Scaling |
| 3 | $S_+ < S_-$ | Ascend | Corrosion |
| 4 | $S_+ > S_-$ | Descend | Scaling |
| 5 | $S_+ > S_-$ | Descend | Scaling |
| 6 | $S_+ > S_-$ | Descend | Scaling |

may impact the accuracy of the model. Second, the so-called actual degradation points in this case are provided by experts based on their experience. In our degradation point detection process, we assume that the calculated value is as close to the real degradation point as possible, but there may be human error from the beginning.

**Table 6. Failure threshold of the primary seal leakage of the main pumps in nuclear power plants.**

| Pump No. | Degradation point | Leakage (l/h) | Change trend | Threshold (l/h) |
|---|---|---|---|---|
| 1 | 1929 | 241 | Ascend | 289 |
| 2 | 2006 | 262 | Ascend | 314 |
| 3 | 636 | 236 | Descend | 197 |
| 4 | 947 | 241 | Ascend | 289 |
| 5 | 452 | 254 | Ascend | 305 |
| 6 | 449 | 258 | Ascend | 310 |

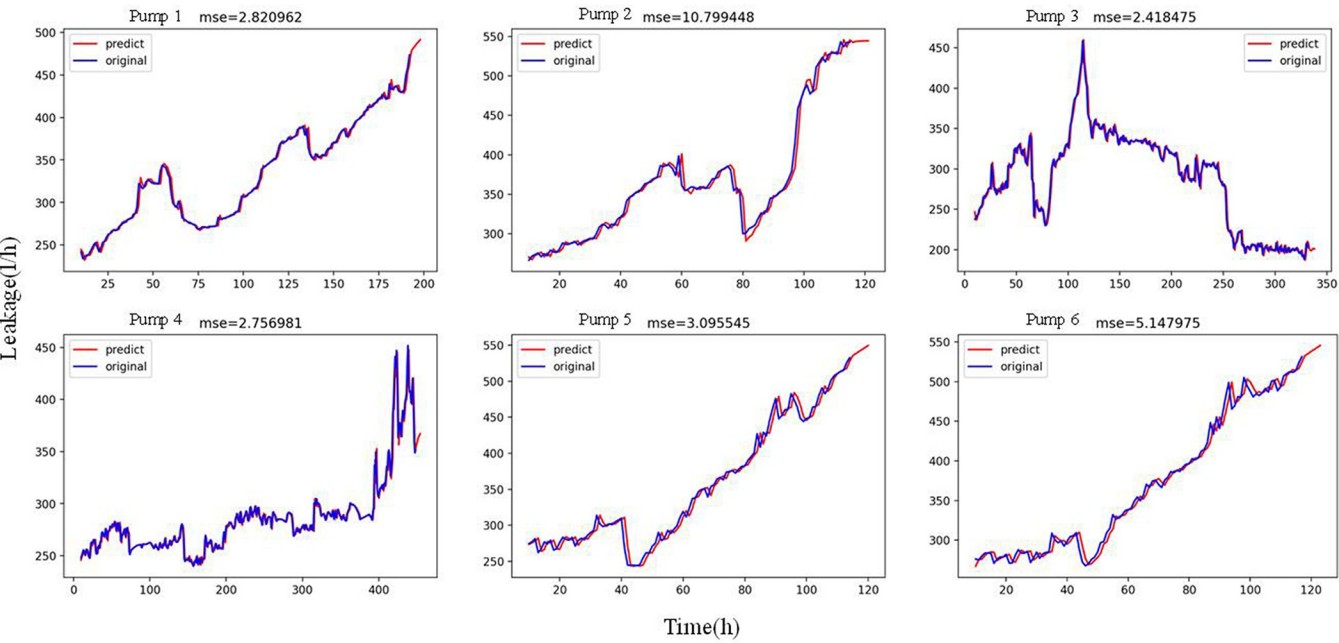

**Fig 11. ARIMA model for short-term prediction.**

In future research, we hope to further integrate this method with the NPP operation and maintenance process, conduct inspection and prediction of the fault diagnosis model, and establish a PHM system for each NPP. Thus, we can achieve real-time fault prediction and health management of NPPs to further improve the safety of NPP operation.

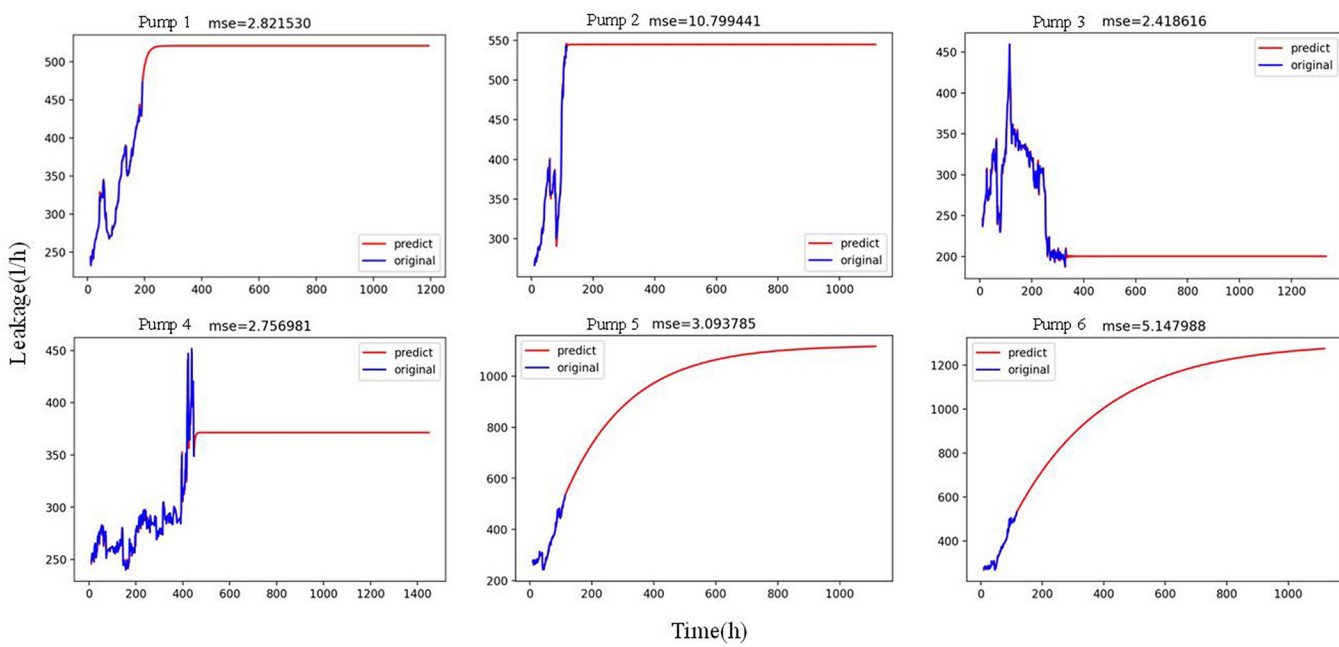

**Fig 12. ARIMA model for long-term prediction.**

**Table 7. Remaining useful life of primary seal leakage of the main pump in a nuclear power plant.**

| Pump No. | Degradation point | Leakage (l/h) | Change trend | Threshold (l/h) | Failure point | RUL (h) |
|---|---|---|---|---|---|---|
| 1 | 1929 | 241 | Rise | 289 | 1971 | 178 |
| 2 | 2006 | 262 | Rise | 314 | 2046 | 170 |
| 3 | 636 | 236 | Descend | 197 | 908 | 1155 |
| 4 | 947 | 241 | Ascend | 289 | 1162 | 913 |
| 5 | 452 | 254 | Ascend | 305 | 485 | 140 |
| 6 | 449 | 258 | Ascend | 310 | 504 | 234 |

## Supporting information

**S1 File. The four indicators monitored from Pump No.1.**
(XLSX)

**S2 File. The four indicators monitored from Pump No.2.**
(XLSX)

**S3 File. The four indicators monitored from Pump No.3.**
(XLSX)

**S4 File. The four indicators monitored from Pump No.4.**
(XLSX)

**S5 File. The four indicators monitored from Pump No.5.**
(XLSX)

**S6 File. The four indicators monitored from Pump No.6.**
(XLSX)

## Acknowledgments

The authors thank Dr. Mohammad Alrwashdeh for helping edit the manuscript, and all the authors for participating in the method discussing and figure illustrating.

## Author Contributions

**Conceptualization:** Yinghua Shao, Rui Kang, Jie Liu.

**Data curation:** Yinghua Shao, Jingwen Fu.

**Formal analysis:** Yinghua Shao, Jie Liu.

**Funding acquisition:** Jie Liu.

**Methodology:** Jie Liu.

**Software:** Jingwen Fu.

**Writing – original draft:** Yinghua Shao.

**Writing – review & editing:** Rui Kang, Jie Liu.

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
