## [Decision Letter · Decision Letter 0]

22 Feb 2024

PONE-D-24-02854An integrated method for leakage fault mode diagnosis and life prediction of the reactor coolant pumpPLOS ONE

Dear Dr. Liu,

Thank you for submitting your manuscript to PLOS ONE. After careful consideration, we feel that it has merit but does not fully meet PLOS ONE’s publication criteria as it currently stands. Therefore, we invite you to submit a revised version of the manuscript that addresses the points raised during the review process.

We look forward to receiving your revised manuscript.

Kind regards,

Dr.Mohammad Alrwashdeh

Academic Editor

PLOS ONE

Journal Requirements:

"Acknowledgement

This work was supported by National Natural Science Foundation of China (No.52005027) and National key Laboratory of Science and Technology on Reliability and Environmental Engineering."

Please state what role the funders took in the study. If the funders had no role, please state: ""The funders had no role in study design, data collection and analysis, decision to publish, or preparation of the manuscript."" If this statement is not correct you must amend it as needed. 

"Acknowledgments

This work was supported by National Natural Science Foundation of China (No.

52005027) (Corresponding author: Jie Liu) and National key Laboratory of Science and

Technology on Reliability and Environmental Engineering."

Please be informed that funding information should not appear in the Acknowledgments section or other areas of your manuscript. We will only publish funding information present in the Funding Statement section of the online submission form. 

"Acknowledgement

This work was supported by National Natural Science Foundation of China (No.52005027) and National key Laboratory of Science and Technology on Reliability and Environmental Engineering."

7. We note that your Data Availability Statement is currently as follows: "All relevant data are within the manuscript and its Supporting Information files."

8. PLOS requires an ORCID iD for the corresponding author in Editorial Manager on papers submitted after December 6th, 2016. Please ensure that you have an ORCID iD and that it is validated in Editorial Manager. To do this, go to ‘Update my Information’ (in the upper left-hand corner of the main menu), and click on the Fetch/Validate link next to the ORCID field. This will take you to the ORCID site and allow you to create a new iD or authenticate a pre-existing iD in Editorial Manager. Please see the following video for instructions on linking an ORCID iD to your Editorial Manager account: https://www.youtube.com/watch?v=_xcclfuvtxQ

9. Please ensure that you refer to Figures 2, 3, and 7 in your text as, if accepted, production will need this reference to link the reader to the figure.

10. We note that Figure 1 in your submission contain copyrighted images. All PLOS content is published under the Creative Commons Attribution License (CC BY 4.0), which means that the manuscript, images, and Supporting Information files will be freely available online, and any third party is permitted to access, download, copy, distribute, and use these materials in any way, even commercially, with proper attribution. For more information, see our copyright guidelines: http://journals.plos.org/plosone/s/licenses-and-copyright.

(1) You may seek permission from the original copyright holder of Figure 1 to publish the content specifically under the CC BY 4.0 license. 

(2) If you are unable to obtain permission from the original copyright holder to publish these figures under the CC BY 4.0 license or if the copyright holder’s requirements are incompatible with the CC BY 4.0 license, please either i) remove the figure or ii) supply a replacement figure that complies with the CC BY 4.0 license. Please check copyright information on all replacement figures and update the figure caption with source information. 

If applicable, please specify in the figure caption text when a figure is similar but not identical to the original image and is therefore for illustrative purposes only.

Reviewers' comments:

Reviewer's Responses to Questions

**Comments to the Author**

1. Is the manuscript technically sound, and do the data support the conclusions?

Reviewer #1: Partly

Reviewer #2: Partly

Reviewer #3: Partly

2. Has the statistical analysis been performed appropriately and rigorously? 

Reviewer #1: I Don't Know

Reviewer #2: Yes

Reviewer #3: Yes

3. Have the authors made all data underlying the findings in their manuscript fully available?

Reviewer #1: Yes

Reviewer #2: Yes

Reviewer #3: No

4. Is the manuscript presented in an intelligible fashion and written in standard English?

Reviewer #1: Yes

Reviewer #2: No

Reviewer #3: No

5. Review Comments to the Author

Reviewer #1: The authors presented new method for the leakage pattern diagnostic in reactor coolant pump. They compared the degradation values obtained from their models with the real degradation data, however there are big errors appeared, the authors requested to explain more their model and the associated results, some other comments are the following

1- All figures need to be modified for better quality, the figures that shown in the manuscript are not good to be presented in term of quality

2- Figure 8 and Figure 9 don’t have legend, what are the bule and orange lines

3- You need to explain more about the results shown in Figure 10. For example, why at n=55 the error is the lowest, and what is the reason of higher errors..etc. please explain your results in more details

4- The results that shown in Table 6 need to be explained in more details.

5- What is the degradation point in more details

6- In page 24, there is a shifting in figure caption

7- It is very difficult to read figure 12 and 13, and many other figures

Reviewer #2: There are too many typos in the paper and the figures are mostly not legible. The procedure is also all based on derived analytical correlations. Please see the marked comments and corrections in the manuscript.

Reviewer #3: - The abstract does not include sufficient quantitative data. Please revise it.

- The Abstract should contain answers to the following questions: What problem was studied and why is it important? What methods were used? What are the important results? What conclusions can be drawn from the results? What is the novelty of the work and where does it go beyond previous efforts in the literature? Please include specific and quantitative results in your Abstract, while ensuring that it is suitable for a broad audience.

- In literature review, the authors should add more recent publications discussed the current used numerical methods to study TH analysis in nuclear power plants' cores such as Computational fluid dynamics (CFD) and other system codes. Why the ARIMA is selected among other codes and numerical methods? Authors are suggested to enhance the literature review with a section regarding the previous used numerical analyses for similar TH studies. Authors are recommended to modify the literature to include this discussion by adding below refs, which are relevant:

• https://doi.org/10.1063/5.0109255

• https://doi.org/10.1155/2019/4375782

• https://doi.org/10.1016/j.ijheatfluidflow.2023.109115

-The originality of the paper needs to be stated clearly. It is of importance to have sufficient results to justify the novelty of a high-quality journal paper. The Introduction should make a compelling case for why the study is useful along with a clear statement of its novelty or originality by providing relevant information and providing answers to basic questions such as: What is already known in the open literature? What is missing (i.e., research gaps)? What needs to be done, why and how? Clear statements of the novelty of the work should also appear briefly in the Abstract and Conclusions sections.

-An updated and complete literature review should be conducted and should appear as part of the Introduction. The results and findings should be compared to and discussed in the context of earlier work in the literature.

- Is the constructed model's predictions validated against previous experimental and/or numerical data? Please explain.

- Conclusion section sound very general. Please support by more quantitative data.

- Please increase the image quality for all figures in the manuscript.

- check the whole manuscript for grammar and typos.

6. PLOS authors have the option to publish the peer review history of their article (what does this mean?). If published, this will include your full peer review and any attached files.

Reviewer #1: No

Reviewer #2: **Yes**

Reviewer #3: No

---

## [Author Response · Author response to Decision Letter 0]

1 Apr 2024

All the reviews have been answered in the letter of "Response to Reviewers".

Thanks for the suggestion from the reviewers.

---

## [Editor Report · Decision Letter 1]

8 Apr 2024

An integrated method for leakage fault mode diagnosis and life prediction of the reactor coolant pump

PONE-D-24-02854R1

Dear Dr. Liu,

We’re pleased to inform you that your manuscript has been judged scientifically suitable for publication and will be formally accepted for publication once it meets all outstanding technical requirements.

Kind regards,

Mohammad Alrwashdeh

Academic Editor

PLOS ONE
---

## [Editor Report · Acceptance letter]

21 May 2024

PONE-D-24-02854R1 

PLOS ONE

Dear Dr. Liu, 

I'm pleased to inform you that your manuscript has been deemed suitable for publication in PLOS ONE. Congratulations! Your manuscript is now being handed over to our production team.

Kind regards, 

on behalf of

Dr. Mohammad Alrwashdeh 

Academic Editor

PLOS ONE